# Lifestyle and Psychological Factors Associated with Pregnancy Intentions: Findings from a Longitudinal Cohort Study of Australian Women

**DOI:** 10.3390/ijerph16245094

**Published:** 2019-12-13

**Authors:** Briony Hill, Mathew Ling, Gita Mishra, Lisa J. Moran, Helena J. Teede, Lauren Bruce, Helen Skouteris

**Affiliations:** 1Monash Centre for Health Research and Implementation, School of Public Health and Preventive Medicine, Monash University, 43-51 Kanooka Grove, Clayton 3168, Australia; lisa.moran@monash.edu (L.J.M.); helena.teede@monash.edu (H.J.T.); lauren.bruce@monash.edu (L.B.); helen.skouteris@monash.edu (H.S.); 2School of Psychology, Deakin University, Locked Bag 20000, Geelong 3220, Australia; m.ling@deakin.edu.au; 3School of Public Health, Faculty of Medicine, University of Queensland, 288 Herston Road, Herston, Brisbane 4006, Australia; g.mishra@sph.uq.edu.au; 4Warwick Business School, Warwick University, Scarman Rd, Coventry CV4 7AL, UK

**Keywords:** preconception, pregnancy intention, lifestyle, health behaviour, psychological well-being

## Abstract

Background: Preconception is a critical time for the establishment of healthy lifestyle behaviours and psychological well-being to reduce adverse maternal and offspring outcomes. This study aimed to explore relationships between preconception lifestyle and psychological factors and prospectively assessed short- (currently trying to conceive) and long-term (future parenthood aspirations) pregnancy intentions. Methods: Data from Wave 3 (age 25–30 years; *n* = 7656) and Wave 5 (age 31–36 years; *n* = 4735) from the Australian Longitudinal Study of Women’s Health were used. Pregnancy intentions and parenthood aspirations were evaluated. Logistic regressions explored cross-sectional associations between demographic, lifestyle and psychological factors and pregnancy intentions/parenthood aspirations. Results: In multivariable models, parity and marital status were associated consistently with pregnancy intentions and parenthood aspirations. Few lifestyle behaviours and no psychological factors were associated with pregnancy intentions. Alcohol intake was the only behaviour associated with aspirations to have a first child. Aspirations for a second/subsequent child were associated negatively with physical activity, sitting time, diet quality, lower anxiety and higher stress. Conclusions: It appears that women are not changing their behaviours when they form a decision to try to conceive. Interventions are needed that address women’s preconception needs, to optimise lifestyle and improve health outcomes for women and their families.

## 1. Introduction

Preconception is a critical time for healthy lifestyle behaviours and positive psychological well-being as these reduce risks for adverse maternal and offspring outcomes during and after pregnancy. Smoking, alcohol intake, diet, and physical activity are all modifiable preconception lifestyle risk factors [1,2,3,4,5,6]. Poorer psychological well-being is associated with poorer lifestyle before and during pregnancy [7,8,9], as well as being a risk factor for postnatal mood disorders and associated complications such as poor child cognitive, physical, and behavioural outcomes [10,11,12].

Pregnancy intentions are an important concept implicated in preconception health [13,14]. Much literature reports independently on the associations between a range of lifestyle and psychological factors with pregnancy intention, yet few studies simultaneously explore both lifestyle and psychological factors and their relationships [13]. Given that lifestyle and psychological well-being are interdependent and influenced by each other [9,15,16], this should considered in analyses. Furthermore, the literature primarily assesses pregnancy intentions retrospectively, potentially introducing bias [17]. The few studies measuring pregnancy intentions prospectively typically report no relationship between pregnancy intentions and smoking, alcohol, physical activity, or diet [18,19,20,21,22,23] and none measured psychological factors. Furthermore, only one study has been conducted outside the U.S. [23].

Consequently, this study sought to address three clear gaps in the literature and provide novel insights into preconception lifestyle and psychological well-being: (1) to explore prospective pregnancy intentions in an Australian cohort; (2) to explore parenthood aspirations as a preconception concept, which to date have been predominantly investigated in younger student samples and have not focused on lifestyle or psychological factors [24,25,26]; and (3) to incorporate multiple psychological factors into a model of modifiable factors associated with prospectively measured pregnancy intentions. Understanding the characteristics of Australian women before pregnancy will contribute to the development of relevant individual and public health strategies to promote health preconception. Therefore, we aimed to investigate the relationship between lifestyle and psychological factors and pregnancy intentions, assessed before pregnancy, in a representative cohort of Australian women. The preconception period can be envisioned from a life course perspective, whereby a woman denotes a conscious intention to conceive (short-term pregnancy intentions), and whereby individuals without immediate pregnancy intentions may also be considered preconception [27]; hence, women with future aspirations for children may be captured here (long-term pregnancy intentions). Using data from the Australian Longitudinal Study of Women’s Health (ALSWH), our specific objectives were to explore the relationships between lifestyle (physical activity, sedentary behaviour, smoking, alcohol use, and diet quality), psychological factors (depression, anxiety, and stress) and short- and long-term pregnancy intentions (i.e., current pregnancy intentions and long-term parenthood aspirations, respectively), while simultaneously accounting for sociodemographic factors.

## 2. Method

This study draws on data from the ALSWH, an ongoing, prospective population-based study following three cohorts of women who were aged 18 to 23, 45 to 50 and 70 to 75 years at enrolment in 1996, with a fourth cohort aged 18 to 23 years enrolled in 2013 [28]. The study examined the health of over 58,000 Australian women. Participants were selected randomly from the national Medicare health insurance database, which includes all Australian citizens and permanent residents. Recruitment methods and the cohort profile have been described previously [28,29,30]. The sample is broadly representative of the general population [29,30]. The ALSWH collected self-reported data via mailed or online surveys. Ethics approval (H-076-0795 and H-2011-0371) was obtained from the Universities of Newcastle and Queensland. Written informed consent was obtained, and access to de-identified data was granted by the data custodians.

### 2.1. Study Population

The current study used data from the ‘younger’ (born 1973–1978) cohort. At baseline (Wave 1; 1996; 18–23 years), 12,432 women completed the survey. For the current study, data from Wave 3 (2003; 25–30 years; *n* = 7656; 61.6% baseline participants) and Wave 5 (2009; 31–36 years; *n* = 4735, 38.1% of baseline participants and 61.8% Wave 3 completers) were included [30]. The impact of attrition has been found to be minimal [31]. Waves 3 and 5 of the ‘younger’ cohort were selected for this study because women were of reproductive age (women aged 20–34 have the highest fertility rate in Australia [32]), and items about pregnancy intentions, parenthood aspirations, behavioural, and psychological variables were available.

Women who reported they were trying to become pregnant but were also using contraception were excluded (Wave 3 *n* = 23; Wave 5 *n* = 99), as were women who were pregnant/postpartum (Wave 3 *n* = 899; Wave 5 *n* = 2299), had a tubal ligation or hysterectomy (Wave 3 *n* = 88; Wave 5 *n* = 285), their partner had a vasectomy (Wave 3 *n* = 161; Wave 5 *n* = 686), or who indicated self-reported fertility issues (i.e., they (Wave 3 *n* = 20; Wave 5 *n* = 132) or their partner (Wave 3 *n* = 12; Wave 5 *n* = 407) were not able to have children). Participants with incomplete food frequency data (>10% items with missing responses) or implausible daily energy intake (>14,700 kJ/day or <2100 kJ/day) were also excluded (Wave 3 *n* = 191; Wave 5 *n* = 111). All other women were included.

### 2.2. Measures

#### 2.2.1. Pregnancy Intentions and Parenthood Aspirations

##### Pregnancy Intentions

At Wave 3, pregnancy intentions were derived from two items exploring contraceptive use. Firstly, women were asked what forms of contraception they use now. Women who responded *none* were asked *which of these best described why you are not using contraception now?* Options included *I am trying to become pregnant* and other reasons such as *I am pregnant* and *I have no male sexual partners now*. At Wave 5, women were asked to respond to the item *I am trying to become pregnant* (*yes*/*no*). Pregnancy intentions were coded as yes/no.

##### Parenthood Aspirations

At Wave 3, women were asked, *when you are 35, would you like to have…no children, 1 child, 2 children, 3 or more children?* Women who had no children and aspired to have no children by age 35 or had already reached the number of children they aspired to have by age 35 were coded as having no future aspirations. Women who reported wishing to have their first child or at least one more child were coded as aspiring for future children. This was further stratified into women who aspired to have their first child (nulliparous) and their second/subsequent child (primiparous, hereafter women aspiring to have ‘another child’).

#### 2.2.2. Demographic and Anthropometric Variables

Information on age, education, marital status, household income, employment status, parity, and country of birth was collected. Self-reported height and weight were used to compute body mass index (BMI; World Health Organization (WHO) classification [33]). All variables were assessed at both Wave 3 and 5, except for country of birth (Wave 1).

#### 2.2.3. Lifestyle Factors

##### Physical Activity

Physical activity was measured via two items from the Active Australia 1999 National Physical Activity Survey [34], which asked women’s frequency and duration of participation in brisk walking, moderate or vigorous leisure activity, and vigorous household or garden chores in the last week. Physical activity was calculated as the sum of the products of total weekly minutes for each domain. The maximum plausible frequency of physical activity bouts per week was set at 56 and the maximum plausible value for duration set at 40 h per week (8 h per day, 5 days per week). Responses were converted to MET (metabolic equivalent) minutes (assigned values of 3, 4, and 7.5 for walking, moderate, and vigorous activities, respectively [35]) and categorised as sedentary (METmins <40), low (METmins 41–600), moderate (METmins 601–1200) and high (METmins ≥1200).

##### Sedentary Behaviour

Sitting (hours per week) was a proxy for sedentary behaviour. Women reported how many hours they usually spend sitting down while doing things like visiting friends, driving, reading, watching television, or working at a desk or computer on a usual weekday and usual weekend day.

##### Smoking

Past and present tobacco use was determined, with responses combined into one smoking variable, dichotomised as non-smoker (never-smoker or ex-smoker) or current smoker.

##### Alcohol Intake

Women were asked how often they usually drink alcohol. Responses were dichotomised as *never* or *any alcohol intake* based on recommendations for alcohol abstinence during the preconception period [36].

##### Dietary Quality

Women completed the Cancer Council Victoria Dietary Questionnaire for Epidemiological Studies (DQES) Version 2, which has been validated in young Australian women [37]. The DQES assesses the frequency of consumption, on average, of 80 food and beverage items during the last 12 months. Response options ranged from *never* to *3 or more times per day*. A diet quality score was derived using the Dietary Guideline Index (DGI) [38], which reflects the Australian Guide to Healthy Eating [39]. However, the alcohol item was modified and coded as 0 (any alcohol) or 10 (no alcohol). The possible range of scores for the DGI was 0 to 130.

#### 2.2.4. Psychological Factors

##### Depressive Symptoms

Depressive symptoms were assessed using the Centre for Epidemiological Studies—Depression Scale shortened version (CES-D 10) [40]. The CES-D 10 assesses frequency of feelings and behaviours during the last week. Responses are scored on a scale from 0 to 3 from *rarely or none of the time* to *most or all of the time*. Summed item response scores range from 0 to 30, with higher scores representing more depressed mood. Consistent with ALSWH approaches, a score of 10 or more was classified as symptoms of probable depression [41]. Cronbach’s alphas for the CES-D 10 were α = 0.563 at Wave 3 and α = 0.575 at Wave 5.

##### Anxiety

Symptoms of anxiety were evaluated using a single item, *In the last 12 months, have you had episodes of intense anxiety (e.g., panic attacks)?* Response options were *never*, *rarely*, *sometimes*, or *often*. This item was treated as an ordinal scale.

##### Stress

Perceived stress was evaluated by the Perceived Stress Questionnaire for Young Women (PSQYW) [42]. The PSQYW, developed for the ALSWH, is internally reliable, unifactorial and has content validity [42]. The scale includes 12 items that assesses stress over the last 12 months in 11 life domains (own health, health of other family members, work/employment, living arrangements, study, money, and relationships with parents, partner/spouse, other family members, girlfriends, and boyfriends). Each item is rated on a 6-point scale ranging from *not applicable*/*not at all stressed* to *extremely stressed*. A mean score was computed (range 0–4); higher scores indicated higher stress. Cronbach’s alphas for the PSQYW were α = 0.685 at Wave 3 and α = 0.714 at Wave 5.

### 2.3. Statistical Analyses

Independent t-tests were conducted for continuous variables and Chi-square test or Fisher’s Exact test for categorical variables to compare characteristics of women with and without pregnancy intentions/parenthood aspirations. The relationships between demographic, lifestyle and psychological factors and pregnancy intentions or aspirations were assessed cross-sectionally at two time points using univariable logistic regression models (IBM SPSS Statistics for Windows, Version 25.0. IBM Corp., Armonk, NY, USA). Then, to control for non-independence, all predictors at each time point were simultaneously included in multivariable logistic regression models. At Wave 3, models were evaluated predicting current pregnancy intention and parenthood aspirations by age 35. At Wave 5, only the model of pregnancy intention was evaluated. Sensitivity analyses were conducted to explore whether self-reported fertility issues impacted the findings.

## 3. Results

### 3.1. Participant Characteristics

Characteristics of women with and without pregnancy intentions and parenthood aspirations at Wave 3 are presented in Table 1. At Wave 3 (25–30 years), the mean age was 27.5 years (SD = 1.5), 29% reported high school only education, 93% were Australian-born, and 57% were married or in a de facto relationship (de facto being the committed relationship of a couple living together). At Wave 3, 7% of women were currently trying to conceive and 90% had future parenthood aspirations (64% for first child and 26% for another child). At Wave 5, 11% of women were currently trying to conceive.

### 3.2. Associations between Pregnancy Intentions and Demographic, Anthropometric, Lifestyle, and Psychological Factors

#### 3.2.1. Pregnancy Intentions

##### Wave 3

At age 25 to 30 years, on univariable analyses, being older, reporting an annual household income of AUD$26,000 to $77,999, being married/de facto, and having an obese BMI were positively associated with pregnancy intention, while participating in paid work, having a tertiary degree, drinking any alcohol, reporting higher anxiety or stress symptoms, and participating in moderate or high levels of physical activity were associated with not having a current pregnancy intention (Table 2). On multivariable analyses, older age, being married/de facto, and obese BMI remained associated with having a pregnancy intention, and having fewer children became significantly associated. Furthermore, having a degree, participating in paid work, and drinking alcohol remained significantly associated with not having a pregnancy intention (Table 2).

##### Wave 5

At age 31 to 36 years, on univariable analyses, reporting an income of $78,000 or above, being married/de facto, and having fewer children were associated significantly with current pregnancy intention. Higher depressive, anxiety, or stress symptoms and being a current smoker were associated with not having a current pregnancy intention (Table 2). On multivariable analyses, having fewer children and being married/de facto remained associated significantly with current pregnancy intention (Table 2).

##### Sensitivity Analyses

Sensitivity analyses exploring whether self-reported fertility issues impacted the findings are shown in Appendix A. Findings remained unchanged with the exception that participating in paid work was associated with not having pregnancy intentions at Wave 5.

#### 3.2.2. Parenthood Aspirations

##### Wave 3

At age 25 to 30 years, on univariable analyses, aspiring for a ‘first child’ was associated with being younger, educated at trade/diploma or formal education level, being in paid work, earning over $26,000, being married/de facto, participating in moderate levels of physical activity, and drinking alcohol. Factors associated with not aspiring for a ‘first child’ were overweight or obese BMI, higher depressive or anxiety symptoms, and smoking (Table 3). On multivariable analyses, younger age, earning at least $78,000, being married/de facto, and consuming alcohol, were associated with aspirations to have a first child, and overweight/obese BMI was associated with absence of aspiration to have a first child (Table 3).

At age 25 to 30 years, on univariable analyses, aspiring for ‘another child’ was associated with being older, married/de facto, and overweight or obese BMI. Factors associated with not aspiring for ‘another child’ were having a trade/diploma or tertiary degree, paid work, income over $78,000, Asian country of birth, spending more time sitting, participating in moderate/high intensity physical activity, poorer diet quality, and lower anxiety symptoms (Table 3). On multivariable analyses, aspiring to have ‘another child’ was associated with being married/de facto, and higher stress and lower anxiety symptoms, while not aspiring to have ‘another child’ was associated with a trade/diploma or degree qualification, paid work, higher income, Asian country of birth, spending more time sitting but also higher levels of physical activity, and poorer diet quality (Table 3).

## 4. Discussion

In this study, we investigated the relationship between lifestyle and psychological factors with prospectively assessed pregnancy intentions and future parenthood aspirations in a large cohort of representative Australian women. Only abstinence from alcohol intake was associated with short-term pregnancy intentions at age 25 to 30 years, and no lifestyle or psychological factors were associated with short-term pregnancy intentions at 31 to 36 years. Any alcohol intake was associated with desiring a first child in the future, while women desiring another child were more likely to be less physically active, spend less time sitting, have poorer diet quality, lower anxiety and higher levels of stress.

Our findings revealed that for women aged 25 to 30, being older, married, obese BMI, fewer children, not in paid work, and lower education level were associated with pregnancy intentions, while for women aged 31 to 36 years, only parity and marital status were significant; these findings are broadly consistent with the literature [18,19,21]. Although demographic factors are not easily modifiable, they could be used to identify women who have pregnancy intentions. Maternal high BMI does represent a concern given adverse maternal and child health implications and offers an opportunity for targeted support and intervention.

We found that there was no association between pregnancy intentions at both age 25 to 30 and 31 to 36 years and diet or physical activity/sedentary behaviours. This is concerning given the importance of achieving optimal diet and physical activity behaviours prior to conception [27]. In particular, while pregnancy is often touted as a “teachable moment” for lifestyle change because women are thought to be motivated for their baby’s health [43], research has shown that the relatively short duration of pregnancy and many competing interests (e.g., fatigue, nausea, social norms, financial concerns) make it very difficult for women to change their behaviour during pregnancy [44]. Moreover, diet and physical activity behaviour change require a long duration to form new habits and these changes should be instigated in the months and even years before pregnancy is planned [27]. While it is possible that the women in this study changed their behaviour before they formed pregnancy intentions, this is unlikely given the similar findings for both Waves 3 and 5 for the short-term pregnancy intentions variable. Clearly there are significant opportunities to explore how we can reach, educate, and motivate women to change their diet and physical activity behaviours before pregnancy.

In our study, abstinence from alcohol when intending to become pregnant (at 25–30 years) is a positive preconception health behaviour, consistent with public health messages on alcohol avoidance with intention to conceive and during pregnancy [36,45], albeit only a small proportion (<10%) of women in our sample reported abstaining from alcohol. In prospective studies, alcohol intake was generally not associated with pregnancy intentions [18,19,20,22]. In the current study, the association between pregnancy intentions and abstinence was not observed at age 31 to 36 years. It is unclear as to why this is the case but may be because Australian women in their thirties are slightly more likely to drink alcohol frequently than women in their twenties (when not considering pregnancy intention) [46]. Future research should explore these relationships further and identify clearer opportunities for risk prevention.

Notably, we did not find an association between smoking status and any of the outcome variables. This is in contrast to an analysis of the same ALSWH cohort at Wave 3 where not smoking was associated with pregnancy intention, but fewer covariates and no psychological factors were accounted for [27]. Additionally, previous findings indicate that smoking is consistently not associated with prospectively assessed unintended pregnancies, supporting our finding [18,21,22]. Australia has low and falling smoking rates, with 25% of women in this cohort smoking at 25 to 30 years, similar to national averages [47]. Despite the relatively low smoking rates, given the strong imperative to cease smoking before conception, our finding that pregnancy intention was not associated with smoking cessation is concerning and suggests opportunities for targeted preconception interventions remain [48].

Psychological factors were not associated with immediate pregnancy intentions in our sample. This was the first study to report on these relationships. However, it is well established that unplanned pregnancies are associated with antenatal depression [49]. Taken together, this may indicate causality, where depressive symptoms are a result of experiencing an unplanned pregnancy. More preconception research is needed to confirm our findings.

Future aspirations to have both a first and another child were associated with several demographic factors. There are few studies providing comparable data. However, two studies assessed longer-term pregnancy planning (more than 12 months in the future), reporting results consistent with the current study for nulliparous women: longer-term pregnancy intentions were associated with younger age, higher income, and marital status [19,21]. Future aspirations to have a first child was also associated negatively with overweight/obese BMI status, aligning with data indicating that first time mothers have lower BMI than multiparous mothers [50]. Recognising demographic and anthropometric factors consistent with long-term parenthood aspirations may help identify individuals requiring counselling for family planning and contraceptive use to prevent unplanned pregnancies.

Our findings also revealed women reporting long-term aspirations for their first child were more likely to drink alcohol, which is consistent with the two comparable studies [19,21]. These findings potentially indicate an opportunity for intervention to promote preconception cessation of alcohol intake, even for women with no immediate pregnancy intentions, due to the risk of unplanned pregnancy. Additionally, future aspirations to have another child was the only outcome associated independently with several lifestyle and psychological variables including physical activity, sitting, diet, and anxiety, albeit the adjusted odds ratio for diet quality was very close to one and its clinical significance could be questioned. Parents report poorer diet and physical activity behaviours than non-parents [51], faced with many barriers to engagement such as time and environmental barriers [52,53,54]. Moreover, the relationship between future parenthood aspirations and these lifestyle and psychological factors may be reflective of societal norms before beginning a family [55]. Whilst there is a scarcity of literature exploring future parenthood aspirations, particularly among adults, one Australian study investigated factors associated with pregnant and postpartum women’s childbearing desires [24]. These factors included financial security, partner stability and willingness, interest in motherhood, living standards, and social concerns, albeit health-related lifestyle behaviours and psychological factors were not investigated. The study suggested that women may strive to achieve a perceived level of “lifestyle” before they consider becoming pregnant. This concept deserves further research attention.

The psychological factors associated with aspirations for future children also align with the early parenting years; having more children is associated with greater levels of stress [56]. Furthermore, women experiencing anxiety symptoms may be less likely to want another child and instead focus on their own mental health. However, to our knowledge, no comparable literature exists. The poorer lifestyle and psychological factors linked to women desiring another child highlight the need to target women in the postpartum period and between conceptions as a preconception opportunity to assist with positive behaviour change and promote well-being.

### Strengths and Limitations

Limitations include bias in self-report measures of lifestyle behaviours, albeit self-report measures are reasonable in large-scale epidemiological studies [57,58]. Secondly, we were not able to assess other health behaviours that impact pregnancy outcomes, such as folic acid supplementation. Thirdly, the Cronbach’s alphas for the CES-D scale were less than optimal and hence findings for depression should be interpreted with caution. Fourth, the single item measure of pregnancy intention was also a limitation. While prospective assessment (a strength of the study) overcomes many limitations associated with retrospective assessment [17], this single-item measure may not comprehensively capture the pregnancy planning process. Additional strengths include the representativeness of the ALSWH sample and inclusion of demographic, lifestyle and psychological covariates, highlighting the unique contribution of the significant predictors in the multivariable model. Given that healthy lifestyle behaviours tend to cluster together and with positive psychological well-being [59], future research should explore potential clusters of modifiable factors and how they are associated with pregnancy planning.

## 5. Conclusions and Future Directions

Overall, several key demographic factors including age, parity, and marital status were associated consistently with pregnancy intentions and aspirations for future children. However, few lifestyle behaviours and no psychological factors were associated with current pregnancy intentions in women in their reproductive prime, or for women who aspired to have their first child in the future. In contrast, parous women with aspirations to have another child reported poorer lifestyle behaviours and psychological well-being than women without these aspirations. Together, the findings suggest that it is the life phase that is most strongly predictive of pregnancy intentions and aspirations, and that women are not generally improving their lifestyle behaviours when trying to conceive. Future research should explore clustering relationships between lifestyle and psychological factors in association with pregnancy intentions. Additionally, future interventions should address women’s preconception needs, both soon before conception for pregnancy planners and with a longer-term approach for women with future parenthood aspirations, with specific attention to the inter-conception phase. Given the WHO recommendation for improving the health of women before pregnancy to better health outcomes for women and their families [60], the preconception period needs to be targeted to optimise modifiable lifestyle behaviours.

## Figures and Tables

**Table 1 ijerph-16-05094-t001:** Characteristics of women with and without pregnancy intentions and parenthood aspirations at age 25 to 30 years (Wave 3).

Variable	All N = 7656		Pregnancy Intention N = 516		No Pregnancy Intention N = 6644		*p*-Value ^	Parenthood Aspiration—First Child N = 4877		Parenthood Aspiration—Another Child N = 1998		No Parenthood Aspiration N = 781		*p*-Value ^
	Value *n* (%) or Mean (SD)	N	Value *n* (%) or Mean (SD)	N	Value *n* (%) or Mean (SD)	N		Value *n* (%) or Mean (SD)	N	Value *n* (%) or Mean (SD)	N	Value *n* (%) or Mean (SD)	N	
*Age, years, mean (SD)*	27.5 (1.5)	7656	27.8 (1.5)	516	27.5 (1.5)	6644	**<0.001**	27.3 (1.4)	4877	28.0 (1.4)	1998	27.6 (1.5)	781	**<0.001** ^†^ **<0.001** ^‡^
*Highest level of education, n (%)*		7482		502		6501	**<0.001**		4789		1928		765	**<0.001** ^†^ **<0.001** ^‡^
No formal education/high school	2148 (28.7)		177 (35.3)		1807 (27.8)			958 (20.0)		964 (50.0)		226 (29.5)		
Trade/diploma	1879 (25.1)		155 (30.9)		1579 (24.3)			1129 (23.6)		568 (29.5)		182 (23.8)		
Degree	3455 (46.2)		170 (33.9)		3115 (47.9)			2702 (56.4)		396 (20.5)		357 (46.7)		
*Country of birth, n (%)*		7594		514		6589	0.338		4834		1983		777	0.456 ^†^ **0.002** ^‡^
Australia	7037 (92.7)		485 (94.4)		6104 (92.6)			4455 (92.2)		1868 (94.2)		714 (91.9)		
Other English-speaking background	281 (3.7)		17 (3.3)		249 (3.8)			180 (3.7)		73 (3.7)		28 (3.6)		
Europe, Asia or Other	276 (3.6)		12 (2.4)		236 (3.6)			199 (4.1)		42 (2.1)		35 (4.5)		
*Formal marital status, n (%) **		7627		514		6620	**<0.001**		4862		1991		774	**<0.001** ^†^ **<0.001** ^‡^
Not married or de facto	3281 (43.0)		18 (3.5)		3052 (46.1)			2429 (50.0)		356 (17.9)		496 (64.1)		
Married or de facto	4346 (57.0)		496 (96.5)		3568 (53.9)			2433 (50.0)		1635 (82.1)		278 (35.9)		
*Annual household income, AUD, n (%)*		5929		466		5095	0.100		3701		1710		518	**<0.001** ^†^ **<0.001** ^‡^
<$26,000	535 (9.0)		30 (6.4)		464 (9.1)			209 (5.6)		270 (15.8)		56 (10.8)		
$26,000 to $77,999	3303 (55.7)		274 (58.8)		2807 (55.1)			1810 (48.9)		1196 (69.9)		297 (57.3)		
≥$78,000	2091 (35.3)		162 (34.8)		1824 (35.8)			1682 (45.4)		244 (14.3)		165 (31.9)		
*Employment status, n (%) **		7656		516		6644	**0.015**		4877		1998		781	**<0.001** ^†^ **<0.001** ^‡^
No paid work	1516 (19.8)		122 (23.6)		1274 (19.2)			514 (10.5)		866 (43.3)		136 (17.4)		
Paid work	6140 (80.2)		394 (76.4)		5370 (80.8)			4363 (89.5)		1132 (56.7)		645 (82.6)		
*Number of children, n (%)*		7656		516		6644	**<0.001**		4877		1998		781	**<0.001** ^†^ **<0.001** ^‡^
Zero	5587 (73.0)		342 (66.3)		4897 (73.7)			4877 (100)		0 (0.0)		710 (90.9)		
One	1033 (13.5)		138 (26.7)		829 (12.5)			0 (0.0)		1014 (50.8)		19 (2.4)		
Two	772 (13.5)		29 (5.6)		687 (10.3)			0 (0.0)		738 (36.9)		34 (4.4)		
Three or more	264 (3.5)		7 (1.4)		231 (3.5)			0 (0.0)		207 (10.4)		18 (2.3)		
*BMI (kg/m^2^), mean (SD)*	24.7 (5.5)	7367	26.0 (6.5)	493	24.6 (5.3)	6402	**<0.001**	24.2 (5.2)	4733	25.9 (6.0)	1892	25.0 (6.1)	742	**<0.001** ^†^ **<0.001** ^‡^
*BMI category, n (%)*		7367		493		6402	**<0.001**		4733		1892		742	**<0.001** ^†^ **<0.001** ^‡^
Underweight	348 (4.7)		18 (3.7)		298 (4.7)			224 (4.7)		84 (4.4)		40 (5.4)		
Normal weight	4322 (58.7)		257 (52.1)		3802 (59.4)			2996 (63.3)		914 (48.3)		412 (55.5)		
Overweight	1588 (21.6)		100 (20.3)		1400 (21.9)			940 (19.9)		481 (25.4)		167 (22.5)		
Obese	1109 (15.1)		118 (23.9)		902 (14.1)			573 (12.1)		413 (21.8)		123 (16.6)		
*Physical activity (METmins), mean (SD)*	1105.2 (1326.4)	7599	918 (1131.6)	510	1118 (1313.4)	6601	**<0.001**	1200 (1351.0)	4842	801 (1055.7)	1984	1291 (1636.3)	773	0.143 ^†^ **<0.001** ^‡^
*Physical activity categories, n (%)*		7599		510		6601	**<0.001**		4842		1984		773	0.166 ^†^ **<0.001** ^‡^
Sedentary	632 (8.3)		58 (11.4)		523 (7.9)			305 (6.3)		265 (13.4)		62 (8.0)		
Low PA	2858 (37.6)		210 (41.2)		2460 (37.3)			1672 (34.5)		918 (46.3)		268 (34.7)		
Moderate PA	1749 (23.0)		114 (22.4)		1531 (23.2)			1167 (24.1)		416 (21.0)		166 (21.5)		
High PA	2360 (31.1)		128 (25.1)		2087 (31.6)			1698 (35.1)		385 (19.4)		277 (35.8)		
*Sedentary behaviour (sitting time), hours, mean (SD)*	6.3 (2.8)	7228	6.3 (2.7)	486	6.3 (2.8)	6399	0.851	6.7 (2.7)	4629	5.1 (2.5)	1881	6.8 (2.9)	718	0.521 ^†^ **<0.001** ^‡^
*Diet quality score, mean (SD)*	77.0 (11.5)	7652	76.8 (11.5)	516	77.1 (11.5)	6642	0.539	77.7 (11.2)	4876	75.1 (11.8)	1996	77.9 (12.2)	780	0.653 ^†^ **<0.001** ^‡^
*Alcohol intake, n (%) **		7638		512		6633	**0.008**		4867		1992		779	**<0.001**^†^ 0.646 ^‡^
None	548 (7.2)		50 (9.8)		437 (6.6)			254 (5.2)		208 (10.4)		86 (11.0)		
Any	7090 (92.8)		462 (90.2)		6196 (93.4)			4613 (94.8)		1784 (89.6)		693 (89.0)		
*Smoking, n (%) **		7624		513		6619	0.491		4855		1989		780	**0.018**^†^ 0.650 ^‡^
Never or ex-smoker	5685 (74.6)		392 (76.4)		4960 (74.9)			3700 (76.2)		1421 (71.4)		564 (72.3)		
Current smoker	1939 (25.4)		121 (23.6)		1659 (25.1)			1155 (23.8)		568 (28.6)		216 (27.7)		
*Depressive symptoms score, mean (SD)*	6.9 (5.3)	7512	6.3 (4.8)	505	6.9 (5.3)	6531	**0.006**	6.7 (5.1)	4793	7.5 (5.4)	1952	7.4 (5.4)	767	**<0.001**^†^ 0.503 ^‡^
*Depressive symptoms category, n (%) **		7512		505		6531	0.152		4793		1952		767	**0.017**^†^ 0.494 ^‡^
No	5572 (74.2)		391 (77.4)		4861 (74.4)			3641 (76.0)		1379 (70.6)		552 (72.0)		
Yes	1940 (25.8)		114 (22.6)		1670 (25.6)			1152 (24.0)		573 (29.4)		215 (28.0)		
*Anxiety symptoms, mean (SD)*	1.3 (0.7)	7638	1.3 (0.6)	512	1.3 (0.7)	6631	**0.030**	1.3 (0.7)	4866	1.3 (0.7)	1993	1.4 (0.8)	779	**0.008** ^†^ **0.003** ^‡^
*Stress, mean (SD)*	0.9 (0.5)	7634	0.8 (0.5)	513	0.9 (0.5)	6630	**0.003**	0.9 (0.5)	4864	1.0 (0.6)	1996	0.9 (0.5)	774	0.225 ^†^ 0.146 ^‡^

^ Comparing with and without pregnancy intention at Wave 3. Note. Data were analysed by independent t-test to compare continuous variables and Chi-square test or Fisher’s Exact test (*) to compare categorical variables between women with and without pregnancy intentions. ^†^ Aspirations to have first child vs. no parenthood aspirations. ^‡^ Aspirations to have another child vs. no parenthood aspirations. Significant values are indicated in bold.

**Table 2 ijerph-16-05094-t002:** Odds ratios (ORs), adjusted odds ratios (aORs), 95% Confidence Intervals (95%CIs), and *p*-values from univariable and multivariable logistic regression analyses highlighting associations between pregnancy intentions and demographic, lifestyle and psychological variables at age 25 to 30 years (Wave 3) and 31 to 36 years (Wave 5).

Variable	Wave 3				Wave 5			
	Univariable		Multivariable		Univariable		Multivariable	
	OR (95%CI)	*p*-Value	aOR * (95%CI)	*p*-Value	OR (95% CI)	*p*-Value	aOR * (95% CI)	*p*-Value
*Age*	1.2 (1.1–1.2)	**<0.001**	1.2 (1.1–1.2)	**<0.001**	1.0 (0.9–1.0)	0.119	1.0 (1.0–1.1)	0.313
*Number of children*	1.0 (0.9–1.1)	0.612	0.5 (0.4–0.6)	**<0.001**	0.7 (0.6–0.7)	**<0.001**	0.4 (0.4–0.5)	**<0.001**
*Education*								
No formal/high school	REF		REF		REF		REF	
Trade/diploma	1.0 (0.8–1.3)	0.985	1.0 (0.7–1.2)	0.425	1.2 (0.9–1.6)	0.144	1.3 (0.9–1.8)	0.202
Degree	0.6 (0.4–0.7)	**<0.001**	0.5 (0.4–0.7)	**<0.001**	1.2 (1.0–1.6)	0.104	1.1 (0.8–1.5)	0.657
*Employment status*								
No paid work	REF		REF		REF		REF	
Paid work	0.8 (0.6–1.0)	**0.014**	0.7 (0.5–1.0)	**0.023**	1.1 (0.8–1.4)	0.617	0.8 (0.6–1.0)	0.106
*Annual household income (AUD$)*								
<$25,999	REF		REF		REF		REF	
$26,000–$77,999	1.5 (1.0–2.2)	**0.038**	1.0 (0.6–1.6)	0.989	1.3 (0.7–2.6)	0.408	0.9 (0.4–2.0)	0.792
≥$78,000	1.4 (0.9–2.1)	0.122	0.9 (0.5–1.4)	0.548	3.0 (1.6–5.8)	**0.001**	1.1 (0.5–2.3)	0.889
*Marital Status*								
Not married/de facto	REF		REF		REF		REF	
Married/de facto	23.6 (14.7–37.8)	**<0.001**	23.4 (12.7–43.1)	**<0.001**	16.8 (10.5–27.0)	**<0.001**	26.8 (14.4–50.1)	**<0.001**
*Country of birth*								
Australia	REF		REF		REF		REF	
Other English-speaking background	0.9 (0.5–1.4)	0.552	0.9 (0.5–1.6)	0.638	1.2 (0.8–2.0)	0.340	1.1 (0.7–1.9)	0.638
Europe	0.2 (0.0–1.4)	0.100	0.3 (0.0–2.1)	0.222	1.9 (0.9–4.0)	0.088	1.8 (0.8–4.5)	0.178
Asia	0.9 (0.5–1.8)	0.831	1.2 (0.5–3.1)	0.739	0.6 (0.2–1.5)	0.304	0.4 (0.1–1.5)	0.192
Other	0.5 (0.1–2.2)	0.372	0.5 (0.1–4.0)	0.536	0.6 (0.1–2.3)	0.425	1.2 (0.3–5.9)	0.787
*BMI category*								
Underweight	0.9 (0.5–1.5)	0.654	0.8 (0.4–1.6)	0.548	0.9 (0.5–1.7)	0.769	1.3 (0.6–2.6)	0.523
Normal weight	REF		REF		REF		REF	
Overweight	1.1 (0.8–1.3)	0.651	1.2 (0.9–1.5)	0.306	0.9 (0.7–1.1)	0.183	0.9 (0.7–1.2)	0.412
Obese	1.9 (1.5–2.4)	**<0.001**	1.7 (1.3–2.3)	**<0.001**	0.8 (0.6–1.0)	0.102	1.3 (1.0–1.8)	0.092
*Physical activity*								
Sedentary	REF		REF		REF		REF	
Low PA	0.8 (0.6–1.10)	0.093	0.8 (0.6–1.2)	0.402	1.2 (0.9–1.7)	0.198	1.0 (0.7–1.5)	0.936
Moderate PA	0.7 (0.5–0.9)	**0.018**	0.8 (0.5–1.2)	0.312	1.2 (0.9–1.7)	0.198	0.9 (0.6–1.4)	0.790
High PA	0.6 (0.4–0.8)	**<0.001**	0.8 (0.5–1.2)	0.350	1.0 (0.7–1.4)	0.915	0.8 (0.5–1.2)	0.200
*Sedentary behaviour (sitting time)*	1.0 (1.0–1.0)	0.954	1.0 (0.9–1.0)	0.319	1.0 (1.0–1.0)	0.431	1.0 (0.9–1.0)	0.065
*Diet quality*	1.0 (1.0–1.0)	0.539	1.0 (1.0–1.0)	0.938	1.0 (1.0–1.0)	0.197	1.0 (1.0–1.0)	0.638
*Alcohol intake*								
None	REF		REF		REF		REF	
Any	0.7 (0.5–0.9)	**0.006**	0.7 (0.4–1.0)	**0.038**	1.1 (0.8–1.5)	0.736	0.8 (0.5–1.2)	0.266
*Smoking*								
Never or ex-smoker	REF		REF		REF		REF	
Current smoker	0.9 (0.7–1.1)	0.456	1.2 (0.9–1.5)	0.223	0.6 (0.5–0.8)	**0.001**	0.7 (0.5–1.0)	0.058
*Depressive symptoms*								
No	REF		REF		REF		REF	
Yes	0.8 (0.7–1.1)	0.136	0.9 (0.7–1.2)	0.571	0.7 (0.6–0.9)	**0.005**	1.1 (0.8–1.5)	0.523
*Anxiety symptoms*	0.9 (0.8–1.0)	**0.046**	0.9 (0.7–1.0)	0.130	0.8 (0.7–1.0)	**0.016**	0.9 (0.8–1.1)	0.436
*Stress*	0.8 (0.6–0.9)	**0.004**	1.1 (0.8–1.4)	0.653	0.7 (0.6–0.8)	**<0.001**	0.8 (0.6–1.0)	0.084

* Multivariable analysis adjusted for all other variables in the table. Note. REF indicates reference category. Significant associations are indicated in bold.

**Table 3 ijerph-16-05094-t003:** Odds ratios (ORs), adjusted odds ratios (aORs), 95% Confidence Intervals (95%CIs), and *p*-values from univariable and multivariable logistic regression analyses highlighting associations between parenthood aspirations and demographic, lifestyle and psychological variables at age 25 to 30 years (Wave 3).

Variable	First Child ^†^				Another Child ^†^			
	Univariable		Multivariable		Univariable		Multivariable	
	OR (95%CI)	*p*-Value	aOR * (95%CI)	*p*-Value	OR (95% CI)	*p*-Value	aOR * (95% CI)	*p*-Value
*Age*	0.9 (0.3–0.9)	**<0.001**	0.8 (0.8–0.9)	**<0.001**	1.2 (1.1–1.3)	**<0.001**	1.1 (1.0–1.2)	0.086
*Education*								
No formal/high school	REF		REF		REF		REF	
Trade/diploma	1.5 (1.2–1.8)	**<0.001**	1.1 (0.8–1.5)	0.608	0.7 (0.6–0.9)	0.006	0.6 (0.4–0.9)	0.005
Degree	1.8 (1.5–2.1)	**<0.001**	1.3 (1.0–1.7)	0.051	0.3 (0.2–0.3)	<0.001	0.3 (0.2–0.5)	<0.001
*Employment status*								
No paid work	REF		REF		REF		REF	
Paid work	1.8 (1.5–2.2)	**<0.001**	1.2 (1.1–1.7)	0.270	0.3 (0.2–0.3)	**<0.001**	0.3 (0.2–0.5)	**<0.001**
*Annual household income (AUD$)*								
<$25,999	REF		REF		REF		REF	
$26,000–$77,999	1.6 (1.2–2.2)	**0.003**	1.1 (0.8–1.7)	0.429	0.8 (0.6–1.1)	0.262	0.4 (0.2–0.7)	**<0.001**
≥$78,000	2.7 (2.0–3.8)	**<0.001**	1.7 (1.1–2.5)	**0.013**	0.3 (0.2–0.4)	**<0.001**	0.2 (0.1–0.4)	**<0.001**
*Marital Status*								
Not married/de facto	REF		REF		REF		REF	
Married/de facto	1.8 (1.5–2.1)	**<0.001**	1.7 (1.4–2.1)	**<0.001**	8.2 (6.8–9.9)	**<0.001**	12.8 (9.1–18.1)	**<0.001**
*Country of birth*								
Australia	REF		REF		REF		REF	
Other English-speaking background	1.0 (0.7–1.5)	0.885	1.0 (0.6–1.7)	0.956	1.0 (0.6–1.6)	0.998	1.2 (0.6–2.4)	0.688
Europe	1.5 (0.7–3.6)	0.309	1.7 (0.5–5.5)	0.411	0.5 (0.2–1.5)	0.214	0.2 (0.0–1.5)	0.109
Asia	0.7 (0.4–1.1)	0.122	0.7 (0.3–1.2)	0.187	0.3 (0.2–0.6)	<0.001	0.3 (0.1–1.0)	0.048
Other	1.2 (0.5–3.0)	0.765	1.7 (0.4–7.4)	0.493	1.1 (0.4–3.0)	0.897	2.3 (0.3–17.1)	0.400
*BMI category*								
Underweight	0.8 (0.5–1.1)	0.146	1.1 (0.6–1.7)	0.837	0.9 (0.6–1.4)	0.785	0.8 (0.4–1.5)	0.446
Normal weight	REF		REF		REF		REF	
Overweight	0.8 (0.6–0.9)	**0.010**	0.8 (0.6–1.0)	**0.030**	1.3 (1.1–1.6)	**0.015**	1.1 (0.8–1.6)	0.491
Obese	0.6 (0.5–0.8)	**<0.001**	0.7 (0.5–0.9)	**0.003**	1.5 (1.2–1.9)	**<0.001**	1.2 (0.8–1.7)	0.341
*Physical activity*								
Sedentary	REF		REF		REF		REF	
Low PA	1.3 (0.9–1.7)	0.123	1.0 (0.6–1.5)	0.907	0.8 (0.6–1.1)	0.159	0.8 (0.5–1.4)	0.430
Moderate PA	1.4 (1.0–2.0)	**0.028**	1.0 (0.7–1.6)	0.848	0.6 (0.4–0.8)	**0.002**	0.6 (0.3–1.0)	**0.050**
High PA	1.2 (0.9–1.7)	0.152	0.9 (0.6–1.4)	0.600	0.3 (0.2–0.4)	**<0.001**	0.5 (0.3–0.8)	**0.005**
*Sedentary behaviour (sitting time)*	1.0 (1.0–1.0)	0.521	1.0 (0.9–1.0)	0.078	0.8 (0.8–0.8)	**<0.001**	0.8 (0.7–0.8)	**<0.001**
Diet quality	1.0 (1.0–1.0)	0.635	1.0 (1.0–1.0)	0.978	1.0 (1.0–1.0)	**<0.001**	1.0 (1.0–1.0)	**0.008**
Alcohol intake								
None	REF		REF		REF		REF	
Any	2.3 (1.7–2.9)	**<0.001**	2.1 (1.5–3.1)	**<0.001**	1.1 (0.8–1.4)	0.646	0.9 (0.5–1.4)	0.533
*Smoking*								
Never or ex-smoker	REF		REF		REF		REF	
Current smoker	0.8 (0.7–1.0)	**0.019**	0.9 (0.7–1.1)	0.256	1.0 (0.9–1.3)	0.650	0.8 (0.6–1.1)	0.180
*Depressive symptoms*								
No	REF		REF		REF		REF	
Yes	0.8 (0.7–1.0)	**0.017**	1.0 (0.8–1.3)	0.810	1.1 (0.9–1.3)	0.494	1.1 (0.8–1.6)	0.522
*Anxiety symptoms*	0.9 (0.8–1.0)	**0.004**	0.9 (0.8–1.1)	0.352	0.8 (0.8–0.9)	**0.002**	0.7 (0.6–0.9)	**0.002**
*Stress*	0.9 (0.8–1.1)	0.225	1.0 (0.8–1.2)	0.694	1.1 (1.0–1.3)	0.147	1.7 (1.3–2.3)	**0.001**

^†^ Comparison group is women with no parenthood aspirations. * Multivariable analyses adjusted for all other variables in the table. Note. Number of children (parity) has not been included as the outcome is based on whether women are aspiring for their first or another child. REF indicates reference category. Significant associations are indicated in bold.

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
