# Peer review of "Lifestyle and Psychological Factors Associated with Pregnancy Intentions: Findings from a Longitudinal Cohort Study of Australian Women"

_ijerph, 2019, doi:10.3390/ijerph16245094_

Round 1

Reviewer 1 Report

Line 52-69: the 3rd objective should be the 1st. Also, the study results should follow the order of the study objectives. Authors should firstly describe pregnancy intentions and parenthood aspirations in Australian women and, subsequently, they should identify correlates for these intentions and aspirations.

Line 260: Descriptive statistics for the variables in which the groups were compared in Table 3. The groups should be specified: women planning their first child and women planning another child were compared with women with no parenthood aspirations.

Reviewer 2 Report

Introduction:

Paragraph 2: the information is inconsistent (“Much literature reports on the associations between a range of lifestyle and psychological factors with pregnancy intention, yet few studies simultaneously explore these factors and their relationships”). Paragraph 3: the gaps are not presented as such, but rather as aims of the present study. It would be clearer if the authors start by identifying the literature gaps and, subsequently, present the aims of the current study. Also, the third aim should be the first, for the sake of organization: the authors should start by describing pregnancy intentions and parenthood aspirations in Australian women and, subsequently, they should identify correlates for these intentions and aspirations. When presenting the study results, the data should follow the order of the study aims.

Method:

Study population: Inclusion criteria for the study sample should be presented. The number of participants that were excluded based on exclusion criteria should be presented. Measures: Dichotomizing alcohol intake based on a recommendation for pregnancy does not seem consistent to me, considering that pregnant women were excluded from the study. The exclusion criterion regarding dietary quality should be presented in the Study Population subsection. The internal consistency for CES-D 10 and PSQYW should be presented. Statistical analyses: the complete list of analyses should be presented here (some are only mentioned in the tables footnotes). The measure for fertility issues should be described in the Measures subsection.

Results:

The sample size is high, which increases the probability of finding a statistically significant result. As such, in order to assess the magnitude of the group differences, effect sizes should be presented. Table 1: the significance value for annual household income should not be in bold. Regarding number of children, a dichotomous variable (children vs. no children) may be more relevant. Comparisons based on continuous variables (for BMI, physical activity, and depressive symptoms) should be added. I think the column “All” is unnecessary. Pregnancy intentions – Wave 3: the results should be presented more rigorously (e.g., the comparison was not between those who reported or did not report anxiety/stress symptoms; as such, instead of “reporting anxiety or stress symptoms”, the authors should state “reporting higher anxiety or stress symptoms”). Table 3: Descriptive statistics are missing for the variables in which the groups were compared. The groups should be specified: women planning their first child and women planning another child were compared with women with no parenthood aspirations, correct?

Discussion:

Paragraph 2: for women aged 31-36, paid work was not associated with pregnancy intentions. Paragraph 3: it should be stressed that, according to Table 1, most women with pregnancy intentions did not abstain from consuming alcohol. Regarding pregnancy intentions, a comment on the results related to the remaining lifestyle factors (diet and physical activity) should be added.
